# Prediction of Drug Stability Using Deep Learning Approach: Case Study of Esomeprazole 40 mg Freeze-Dried Powder for Solution

**DOI:** 10.3390/pharmaceutics13060829

**Published:** 2021-06-03

**Authors:** Jovana Ajdarić, Svetlana Ibrić, Aleksandar Pavlović, Ljubiša Ignjatović, Branka Ivković

**Affiliations:** 1Department of Pharmaceutical Technology and Cosmetology, Faculty of Pharmacy, University of Belgrade, Vojvode Stepe 450, 11221 Belgrade, Serbia; svetlana.ibric@pharmacy.bg.ac.rs (S.I.); branka.ivkovic@pharmacy.bg.ac.rs (B.I.); 2Faculty of Physical Chemistry, University of Belgrade, Studentski trg 12-16, 11158 Belgrade, Serbia; aleksandar.pavlovic@hemofarm.com (A.P.); ljignjatovic@ffh.bg.ac.rs (L.I.)

**Keywords:** esomeprazole sodium, stability, deep learning, multilayer perceptron, artificial neural network, freeze-drying, pH, temperature, related substances, assay

## Abstract

A critical step in the production of Esomeprazole powder for solution is a period between the filling process and lyophilization, where all vials, partially closed, are completely exposed to environmental influences. Excessive instability reflects in pH value variations caused by oxygen’s impact. In order to provide pH control, which consequently affects drug stability, Esomeprazole batches, produced in the same way, were kept in partially closed vials for 3 h at temperatures of 20 °C and −30 °C, after which they were lyophilized and stored for long-term stability for 36 months. The aim of the presented study was to apply a deep-learning algorithm for the prediction of the Esomeprazole stability profile and to determine the pH limit for the reconstituted solution of the final freeze-dried product that would assure a quality product profile over a storage period of 36 months. Multilayer perceptron (MLP) as a deep learning tool, with four layers, was used. The pH value of Esomeprazole solution and time of storage (months) were inputs for the network, while Esomeprazole assay and four main impurities were outputs of the network. In order to keep all related substances and Esomeprazole assay in accordance with specifications for the whole shelf life, the pH value for the reconstituted finish product should be set in the range of 10.4–10.6.

## 1. Introduction

Data mining is recognized as a useful tool in the prediction and control of drug product quality profiles in a Pharma 4.0 concept in the pharmaceutical industry. Deep learning is a subfield of data mining methods that uses algorithms called artificial neural networks (ANNs). It has been successfully used over the past two decades in formulation and process development both in academia and in pharmaceutical applications [1,2,3,4]. The main advantage of ANNs is that it is possible to obtain the prediction of numerous variables at the same time, as well as establish complex models between dependent and independent variables, even in the cases where traditional methods, such as experimental design, are useless. ANNs may contain different numbers of layers, as well as different numbers of neurons in each layer; additionally, the user can choose learning algorithms, activation, and post-synaptic functions; thus, there are numerous possibilities for network architecture [5]. There are only a few attempts in the literature of data mining applications in drug stability prediction. In our previous study, we compared multiple regression analysis (MRA) and dynamic neural network (DNN) for the prediction of the stability of Hydrocortisone 100 mg freeze-dried powder for injection. Degradation products of hydrocortisone sodium succinate were followed during stability studies. All data obtained during stability studies were used for modeling and the superiority of DNN over mathematical modeling was confirmed [6].

Lyophilization or “freeze-drying”, a process that makes a product “water-loving”, provides stability for all formulations that are not stable as a solution, or contain a significant amount of water. The final goal of lyophilization is to provide a final product that is stable, easily reconstituted, and provides acceptable chemical or biological activity for its intended purpose. The lyophilization process includes the preparation of material consisting of the preparation of the solution of active pharmaceutical ingredient (API) and excipients in a suitable solvent—most often water—its freezing, and then the removal of water by a combination of sublimation and desorption [7]. Before they undergo lyophilization, the stability of drug solutions can be affected by: temperature, exposure to light, pH of the solution, presence of additives, concentration of API, presence of oxygen, and duration of storage [8]. Therefore, all these factors should be well known and understood before a single formulation is selected for the production process that includes lyophilization. The freezing step of the freeze-drying process is often the most critical and consequently has received significant attention because this first step of the process governs not only the performance of subsequent steps (primary and secondary drying) but also the product quality (i.e., physical stability, cake appearance, residual moisture, reconstitution time) [9]. The most harmful factors through this freezing process are pH shift, dehydration, increased ionic strength, and formation of interfaces between water and ice [10,11]. The introduction of the ice–aqueous interfaces and pH shifts are well known to cause stability problems. Physical stability is affected by pH, temperature, ionic strength, freeze–thawing, API concentration, and different kinds of mechanical stress [12].

The requisite for the stabilization of the product Esomeprazole 40 mg powder as a solution for injection/ infusion has been triggered by the instability of the drug product solution, which primarily lies in the sensitivity of the API itself to oxygen. API Esomeprazole sodium (presented in Figure 1), a proton pump inhibitor used in the treatment of dyspepsia, peptic ulcer disease (PUD), gastroesophageal reflux disease (GORD/GERD), and Zollinger–Ellison syndrome, with its exceptional chemical sensitivity, makes it necessary to create a clear picture of its impurities and degradation pathways [13,14,15,16]. Esomeprazole sodium is extremely unstable, sensitive to heat and oxidation, and highly susceptible to acid hydrolysis with the appearance of a large number of degradation products. The main degradation products of Esomeprazole occur under thermal and oxidative degradation conditions [16]. Precisely, the chemical properties of API, as well as its therapeutic indications, require providing a stable pharmaceutical form suitable for use in the treatment of these conditions, when the oral route is not possible.

European Pharmacopeia describes nine process-related impurities for Esomeprazole sodium, which could be divided into several groups based on the potential degradation pathways. Process-related impurities (Imp. A, Imp. B, Imp C, Imp. D, Imp. E, Imp. F, Imp. G, Imp. H, Imp. I), degradation products affected by the temperature (Imp. A, Imp. B, Imp C, Imp. D, Imp. E, Imp. F, Imp. G, Imp. H), by acid medium (Imp. C, Imp. D, Imp. F, Imp G, Imp. H), by alkaline medium (Imp. A, Imp. B, Imp. C, Imp. D, Imp. F, Imp. G), by light (Imp. A, Imp. D, Imp. F, Imp. G), and by oxidation (Imp. D, Imp F, Imp. G).

Brändström et al. extensively investigated the chemical reactions of omeprazole and its analogs in the absence and presence of 2-Mercaptoethanol [17]. Brezeziska [18] examined the effect of temperature and relative humidity on the stability of omeprazole in the solid state. Dong et al. (2013) [19] examined the degradation behavior of Esomeprazole by determining the degradation products formed under the influence of acidic, basic, oxidative, photolytic, and thermal stress conditions. Shankar et al. (2019) proved that the drug undergoes degradation under acid, base, neutral hydrolysis, and oxidative degradation conditions forming a total of 16 degradation products, which were characterized by LC-MS/MS experiments and accurate mass measurements [20]. Significant degradation occurs under the influence of acidic and oxidative stress conditions. The previous literature data, as well as the study of forced degradation conducted in our research laboratory, indicated that pH, temperature, and oxygen are key factors for the stability of omeprazole, of which pH plays the most important role.

Baishya et al. (2016) compared the lyophilization process cycle between lab-scale and scale-up batches in order to develop a stable lyophilized injectable dosage of Esomeprazole sodium [21]. The optimal lyophilization process and comparable between scale-up and lab batches were obtained when sublimed temperature setting ensured a sample temperature lower than the collapse temperature of −19.2 °C, estimated by the freeze-drying microscope. In order to develop an optimized lyophilization process for Esomeprazole sodium, in the study of Venna et al. (2013), total cycle time, freezing and holding time, and primary and secondary drying time were varied, while keeping the quantities of all the pharmaceutical ingredients constant [22]. The lyophilization cycle of a total duration of 35 h gave the product with the best results, stable for the prescribed period on the accelerated storage conditions.

The aim of the presented study was to apply a deep-learning algorithm for the prediction of the Esomeprazole stability profile in a freeze-dried product and to determine the limit of the pH value for the reconstituted solution of the final freeze-dried product that would assure a quality product profile over a storage period of 36 months. Multilayer perceptron (MLP) as a deep learning tool, with four layers, was used. The pH value of Esomeprazole solution and time of storage in months were inputs for the network, while Esomeprazole assay and four main impurities were outputs of the network.

## 2. Materials and Methods

### 2.1. Materials

For the solution of the drug product, all raw materials of Ph. Eur. 10.0 quality were used: API Esomeprazole sodium from the producer Union Quimica Farmaceutica, Barcelona, Spain; Disodium edetate as a complexing agent, and pH adjuster Sodium hydroxide from the producer Merck KGaA, Darmstadt, Germany; water for injection (in-house prepared) as a solvent. For the HPLC methods for the analysis of Esomeprazole assay and related substances, the following were used: Glycine, reagent grade; water, HPLC grade; Sodium hydroxide, reagent grade; Acetonitrile, HPLC grade; Methanol, HPLC grade; Disodium hydrogen phosphate, dodecahydrate (Na_2_HPO_4_ × 12H_2_O), reagent grade; Orthophosphoric acid, reagent grade (all reagents from Waters Corporation, Milford, CT, USA).

### 2.2. Preparation of the Solution and the Lyophilization Process

Manufacturing of Esomeprazole 40 mg powder for solution for injection/infusion starts by the preparation of a bulk solution obtained by the simple dissolution of solid, freely soluble raw materials in water for injections (WFI), pH adjustment, and solvent addition until the prescribed final volume. The system for aseptic preparation and first filtration for bioburden reduction consists of a preparation tank, filter unit, and receiving tank (Lurgi TPS AG, Bubendorf, Switzerland). Bulk solution preparation is followed by sterile filtration and filling into vials which are partially closed on the machine Sterifill F200 (IMA, Fairfield, CT, USA), after which lyophilization begins in the freeze dryer Lyomax 16 (BOC Edwards Pharmaceutical Systems, Tonawanda, NY, USA), and at the end, an inspection of fully closed and crimped vials is undertaken (semi-automatic machine for visual inspection V90+, Seidenader, Markt Schwaben, Germany). All production phases are presented in the flow chart of the manufacturing process given in Figure 2, where the review of process and in-process parameters for each production phase is given.

The Esomeprazole solution is produced in the completely closed system, supported by the constant purging of nitrogen, starting from the moment of its preparation until the filling in the appropriate vials. This provides conditions of fully controlled pH, temperature, and oxygen content in the solution, and protects it from all environmental impacts that could potentially jeopardize the product quality. However, the critical step in a production process is a period between the filling process and lyophilization, because in this period all vials containing Esomeprazole solution will be partially closed, and in that way become fully exposed to environmental influences, at the first place of oxygen impact. Excessive instability of Esomeprazole in solution reflects in the variations of the pH value of the solution due to sensitivity to oxygen. This lack of stability further leads to many issues such as changes in related substances’ content, affecting the assay results through the shelf life. Starting from the pH value of the bulk Esomeprazole solution (specification: 10.8–11.8), which decreases at different speeds depending upon the conditions to which it is exposed (temperature, oxygen, and time), makes it hard to control and predict the pH value for the lyophilized product at the end of the production process (specification: 10.0–11.0), and consequently, to control its behavior during its shelf life. Both acceptance criteria, for the pH value of the bulk solution and the pH value of the reconstituted finish product, are estimated as a part of ICH guideline requirements on the specifications for a parenteral drug product, and limits are set according to literature data for the innovator drug product, as well as data gathered from formulation development and stability study results.

In order to provide as best as possible control of the pH value of the solution, which affects the pH value of the finished product, Esomeprazole batches, produced in the same way, were kept in the partially closed vials at room temperature (of about 20 °C) and at the temperature which corresponded to the freezing step of the lyophilization program (−30 °C). These two temperatures were chosen as the only two extreme cases that are possible to occur during the established production process of Esomeprazole 40 mg powder for solution for injection/ infusion based on the environmental conditions in the production area. After a maximum of 3 h of standing in both temperatures, which corresponds to the time duration necessary for the filling of the full batch size, lyophilization started for all examined vials.

From the all produced batches, 5 stored at a temperature of 20 °C and 4 kept at a temperature of −30 °C (presented in Table 1) were chosen to be placed in the stability study at long-term (25 °C/60% RH) and intermediate storage conditions (30 °C/75% RH) for the prescribed shelf life of the product of 36 months. Storage conditions and testing frequency are given in Table 2. Esomeprazole assay, impurities, and pH were tested at every testing time point after conditioning.

Within this study, results for the 132 batches, the total produced, were evaluated. The scheme of trials performed is given in Table 1 below.

### 2.3. pH Measurement

The pH of the solution was potentiometrically measured using a method according to EP 2.2.3 on the device Sartorius^®^ professional meter PP 50 (Sartorius Mechatronics Corp., Edgewood, NY, USA).

### 2.4. Related Substances Test

For the related substances, a determination-modified HPLC method, official according to USP43, was used on the Acquity H-Class^®^ UPLC system (Waters Corporation, Milford, CT, USA). The chromatographic procedure was carried out using Luna^®^ C18, 250 mm × 4.6 mm column packed with particles of silica which was modified by chemically bonded octadecylsilyl groups (5 μm), with a column temperature of 35 °C. As a mobile phase 1, a filtered and degassed mixture of 3.00 g of Glycine dissolved in 1000 mL of water and with pH adjusted at 9.00 ± 0.05 using 50% aqueous Sodium hydroxide was used. As a mobile phase 2, a filtered and degassed mixture of 850 mL of Acetonitrile and 150 mL of Methanol was used. Volume of each injection was 20 µL, with a flow rate of 1.0 mL/min, run time of 40 min, and detector on 280 nm.

### 2.5. Esomeprazole Assay

For the Esomeprazole assay determination, an in-house HPLC method was used on the Acquity H-Class^®^ UPLC sistemu (Waters Corporation, Milford, CT, USA). The chromatographic procedure was carried out using a Luna^®^ C18 (2), 150 mm × 4.6 mm column packed with particles of silica which was modified by chemically bonded octadecylsilyl groups (5 μm). As a mobile phase, a filtered and degassed mixture of 750 mL of buffer pH 7.60 (1.400 g of Disodium hydrogen phosphate, dodecahydrate (Na_2_HPO_4_ × 12H_2_O) dissolved in 1000 mL of water with pH 7.60 ± 0.05 adjusted using 20% aqueous Orthophosphoric acid) and 250 mL of Acetonitrile was used. The volume of each injection was 10 µL, with a flow rate of 2.0 mL/min, and detector on 280 nm.

### 2.6. Artificial Neural Network Modelling

Artificial neural network modeling was applied using software: TIBCO Statistica^®^ Software 13.5.0 (StatSoft Inc., Tulsa, OK, USA). Inputs for the network were the pHs of the reconstituted solution of 9 batches (5 batches stored at a temperature of 20 °C and 4 batches kept at a temperature of −30 °C) and time (in months) during the stability study (0, 3, 6, 9, 12, 18, 24, 30, 36). Five network outputs correspond to Esomeprazole assay and four impurities (namely: 4-Hydroxy Sulphide impurity, Sulphone impurity, 5-Methoxy-1H-benzimidazole-2-sulfinic acid, 4-Hydroxy impurity). Data from 81 experimental runs were divided into training, validation, and test set (64:13:4). A multilayer perceptron (MLP) network was trained using a back-propagation algorithm. The best network was chosen based on the least RMS values for training, validation, and test data set. The validated model was used for establishing the design space for the lyophilization process.

## 3. Results and Discussion

### 3.1. Review of pH Value Influence on the Stability of the Drug Product

Esomeprazole bulk solutions’ pH value is adjusted to the specification values (specification: 10.8–11.8) during solution preparation by the addition of the appropriate quantity of the buffer solution and this is the only production step where pH can be influenced directly by the production process itself. Exposure of the drug product solution to the air increases the content of oxygen in the solution, resulting in a fast-decreasing pH value. As it is explained, this occurs primarily for the solution in the partly closed vials from the moment of their filling until the moment of the lyophilization process start. By this instability, the pH value from the finished product (after reconstitution) is also endangered and cannot be easily controlled (specification: 10.0–11.0). The increased content of oxygen in the bulk solution not only causes a fast decrease of the pH value and consequently leads to results out of specification, but can, respectively, trigger the oxidation of API and lead to an increase of impurities and out-of-specification results for related substances and assay.

Results for the pH of the bulk solution (Figure 3) and pH of the finished product (Figure 4) obtained for the two groups of examined trials are presented.

The results of pH values in the finished product are more shifted toward the upper specification limit for batches kept at −30 °C (average value: 10.69) than for those kept at 20 °C (average value: 10.51). If we observe individual values, the first three batches that were kept at −30 °C show the highest results (10.9, 10.8, 11.0, respectively), while the rest of the batches have comparable results with those recorded for the batches kept at 20 °C. Internal control limits, which were used during the preparation of the solution for all examined batches, in the only production step where pH is directly influenced by the addition of the buffer solution, were set to the narrowed range of 11.8–11.6 in order to exclude inter-individual differences between batches and made all subjected batches comparable. Even though the same “start” point for bulk solutions’ pH of 11.6–11.8 was used, by analyzing all presented pH results from the solution and finished product, it was concluded that a higher pH drop is observed for batches that were kept in partially closed vials at 20 °C:The pH drop for vials at 20 °C was 1.1–1.5 units;The pH drop for vials at −30 °C was 0.8–1.2 units, which is an average of 0.3 less than batches kept at 20 °C before lyophilization.

The pH drop is a consequence of the interaction of the air (oxygen) with the drug product solution. The contact time between oxygen and solution was the same for batches exposed to oxygen at the temperatures of 20 °C and −30 °C, but with the significant difference being that immediately after the solution from partially closed vials was exposed to the freezing temperature it became inert and insensitive to all potential further impacts; that is, the influence of oxygen. This is the reason why these batches have a lower pH drop.

The trend of pH results obtained during stability testing for the batches stored at temperatures of 20 °C and −30 °C is shown in Figure 5:

Comparing the pH results obtained during the shelf life for batches kept at 20 °C, significant differences between testing time points were observed: pH ranges from 10.3 to 11. On the other hand, the pH value for batches kept at −30 °C does not vary significantly during shelf life; pH value ranges from 10.4 to 10.8 at both storage conditions (25 °C/60% RH and 30 °C/75%RH). A more stable pH value during shelf life can be observed for batches that have higher initial pH recorded (for the bulk solution, at the end of the solution-preparation phase).

### 3.2. Review of Stability Study Results

During the stability studies, five potential degradation products were observed with a level that exceeded ICH identification thresholds during storage and were therefore identified. These degradation products are:Sulphone impurity (Impurity D): European Pharmacopoeia Omeprazole Impurity D.4-Hydroxy Sulphone impurity: European Pharmacopoeia Omeprazole Impurity I.4-Hydroxy impurity: product of thermal degradation of Esomeprazole.4-Hydroxy Sulphide impurity: product of hydrolysis in alkaline aqueous solutions (pH 10.5) of 4-Hydroxy Esomeprazole.5-Methoxy-1H-benzimidazole-2-sulfinic acid: hydrolysis product of Omeprazole Impurity D.

#### 3.2.1. Review of Results of Assay and Related Substances during Stability Study

##### Results of Assay of Esomeprazole

The trend of Esomeprazole assay results tested for all tested batches is presented in Figure 6.

All results at both conditions were within specifications. Results usually range from around 98% to 102%. The assay of Esomeprazole does not vary significantly during conditioning. Greater variation is observed at conditions 30 °C/75% RH, but the results are well within specification.

##### Trend of Sulphone Impurity (Impurity D)

The trend of Sulphone impurity results was analyzed for all batches (kept at 20 °C and −30 °C) placed on stability testing (storage condition 25 °C/60% RH and 30 °C/75% RH) and is given in Figure 7.

All results are within specification except for one batch at conditions of 25 °C/60% RH (batch number 5 kept at 20 °C, but produced with API which had a high level of Sulphone impurity, which proceeded its growth during storage). The assay of Sulphone impurity, with the exception of batch 5: 20 °C, stays at the same level at both conditions (usually 0.1%).

##### Trend of 4-Hydroxy Impurity

The trend of 4-hydroxy impurity results is given in Figure 8.

The assay of 4-hydroxy impurity shows a constantly increasing trend during the stability study at both conditions. There is no difference between results at different conditions. By the end of the shelf life for all batches, the assay of this impurity is around 0.5%, far enough from the prescribed specification limit of 0.9%.

##### Trend of 4-Hydroxy-Sulphone Impurity

Results for 4-hydroxy-sulphone impurity obtained during the stability study for all batches is around 0%, or not detected (specification NMT 0.6%) and are presented in Figure 9.

##### Trend of 4-Hydroxy-Sulfide Impurity

The trend of 4-hydroxy-sulfide impurity results obtained during the stability study is given in Figure 10.

The assay of 4-hydroxy sulfide impurity for most of the examined batches stays at the same level during shelf life—around 0.1%. The increase of 4-hidroxy sulfide assay can be noticed in both conditions for batches 2 and 3, which are stored at −30 °C before lyophilization. All results are within specifications (limit 0.4%).

##### Trend of 5-Methoxy-1H-benzimidazole-2-sulfinic Acid

The trend of results for 5-Methoxy-1H-benzimidazole-2-sulfinic acid obtained during stability, for all batches produced (stored at both temperatures of 20 °C and −30 °C before lyophilization), is given in Figure 11.

Results for most of the batches kept at 20 °C and −30 °C before lyophilization started are within specifications except for batch number 5 (kept at 20 °C), batch number 2, and batch number 3 (kept at −30 °C) at both storage conditions of 25 °C/60% RH and 30 °C/75% RH. By comparing the results for batch 2 at different conditions, it can be observed that the assay of 5-Methoxy-1H-benzimidazole-2-sulfinic acid is higher at more extreme conditions of 30 °C/75% RH. At 25 °C/60% RH for most batches, the results usually stay around 0.1%.

From the results obtained at different storage conditions, it can be assumed that temperature has an influence on the formation of 5-Methoxy-1H-benzimidazole-2-sulfinic acid. This is probably related to the influence on the formation of its precursor, impurity D, and more pronounced hydrolysis at a higher temperature. The decrease of Impurity D at 12 months for batches 2: −30 °C and batch 3: −30 °C was also observed, and is probably related to the degradation of Impurity D into 5-Methoxy-1H-benzimidazole-2-sulfinic acid since from this testing point it also starts its rise.

Batches with higher results for 5-Methoxy-1H-benzimidazole-2-sulfinic acid (please see Figure 11) had a higher pH in the final control of the reconstituted solution, such as batch 2: −30 °C, batch 3: −30 °C, and 5: 20 °C (Figure 4), which suggests that pH may have an influence on the assay of this impurity and therefore potentially on product quality. These are also batches for which the pH value for the bulk solution was at the upper limit of specification (around 11.8, given in Figure 3). This assumption was additionally confirmed by the comparison with the results of batch number 15: −30 °C. This batch was produced using the same API batch as batches numbers 2: −30 °C and number 3: −30 °C, kept at the same temperature, but with a lower pH value (10.5) at the finished product and had 0.1% for 5-Methoxy-1H-benzimidazole-2-sulfinic acid during their shelf life (Figure 11). Therefore, it can be concluded that the formation of 5-Methoxy-1H-benzimidazole-2-sulfinic acid is pronounced at a higher pH value and the pH dependency can be assumed. Based on all available results, pH of 10.8 and higher in the finished product control of a reconstituted solution is a trigger for the changes in related substances results.

Differences between batches with a higher assay of 5-Methoxy-1H-benzimidazole-2-sulfinic acid (trials with vials kept at −30 °C, batch number 2: −30 °C, and number 3: −30 °C) and those from the trials at a temperature of 20 °C, e.g., the lower temperature, could only support the stability of API by making the solution inert and not the further degradation into impurity D or 5-Methoxy-1H-benzimidazole-2-sulfinic acid. However, these batches, even with a more stable pH through the shelf life, had a lower pH drop. This again confirms the cause of out-of-specification results for 5-Methoxy-1H-benzimidazole-2-sulfinic acid—a higher pH value of the finished product.

Taking into consideration that the combination of alkaline pH and temperature impact the 5-Methoxy-1H-benzimidazole-2-sulfinic acid impurity occurrence, the evaluation of all pH results available for so far produced batches from both trials was performed in order to find a correlation between the pH value from the final solution and the pH value in the finished product control. The aim was to reveal the design space that describes internal control limits for the pH value of the reconstituted product in the finished product control, which will keep the assay of 5-Methoxy-1H-benzimidazole-2-sulfinic acid, but also for all other impurities, including the assay of Esomeprazole, within specifications during the product shelf life of 36 months.

### 3.3. Artificial Neural Network Modelling

In order to establish an artificial neural network model that would be able to predict the Esomeprazole assay and impurity profile depending on the pH value of the final product, for 36 months of storage, as well as to determine the pH value limit that will assure final product quality over the storage period, a four-layer MLP network (2-3-3-5) was trained using a back-propagation algorithm (Figure 12).

Two input layers correspond to the pH value of the reconstituted product and time (in months) of storage. Outputs were Esomeprazole content and four main impurities. The data set, consisting of 81 runs, was divided into training, validation, and test set (64:14:4). The network was trained through 10,000 epochs. After training, RMS error for training, validation, and test set were 0.025, 0.038, and 0.013, respectively, which correspond to the successful prediction of stability results.

The process conditions of pH range for the finished product (reconstituted freeze-dried product) which will stabilize the Esomeprazole 40 mg powder for solution for injection/infusion in terms of related substances and assay in accordance with specifications during the whole shelf life of 36 months were defined by using the design space obtained from the gathered data.

Figure 13a presents the response surface, generated using the validated neural network, while Figure 13b present a contour graph. Contours are selected according to specification limits for 5-Methoxy-1H-benzimidazole-2-sulfinic acid: ≤0.2%. Areas in dark green are design space for the assay of 5-Methoxy-1H-benzimidazole-2-sulfinic acid, representing the limits of pH value and the prescribed product shelf life of 36 months within the levels that 5-Methoxy-1H-benzimidazole-2-sulfinic acid will be within specification limits.

Based on obtained results, in Figure 13b the design space is presented; namely, the boundaries of pH value of the product (10.4–10.6) in which the assay of 5-Methoxy-1H-benzimidazole-2-sulfinic acid remains stable and in accordance with the acceptance criterion prescribed by the specification (≤0.2%) for the defined shelf life of 36 months.

In Figure 14a the response surface is presented, generated using the validated neural network, while Figure 14b presents the contour graph. Contours are selected according to specification limits for the assay of Esomeprazole 95–105%.

Based on obtained results, in Figure 14b the design space is presented; namely, the boundaries of the pH value of the product (10.2–11.0) in which the assay of Esomeprazole remains stable and in accordance with the acceptance criterion prescribed by the specification for the defined shelf life of 36 months.

In Figure 15a the response surface is presented, generated using the validated neural network, while Figure 15b presents the contour graph. Contours are selected according to the specification limits for the assay of Sulphone impurity: ≤0.4%. All areas beside light green are design spaces for the assay of Sulphone impurity, representing the limits of pH value and the prescribed product shelf life of 36 months within which the levels of assay of Sulphone impurity will be within specification limits.

Based on obtained results, in Figure 15b the design space is presented; namely, the boundaries of the pH values of the product (10.2–10.6), in which the assay of Sulphone impurity remains stable and in accordance with the acceptance criterion prescribed by the specification for the defined shelf life of 36 months.

In Figure 16a the response surface is presented, generated using the validated neural network, while Figure 16b presents the contour graph. Contours are selected according to specification limits for the assay of 4-Hydroxy impurity: ≤0.9%. All areas presented are design spaces for assay of 4-Hydroxy impurity, representing the limits of pH value and the prescribed product shelf life of 36 months within which the levels of assay of 4-Hydroxy impurity will be within specification limits.

Based on obtained results, in Figure 16b the design space is presented; namely, the boundaries of the pH value of the product (10.3–11.1) in which the assay of 4-Hydroxy impurity remains stable and in accordance with the acceptance criterion prescribed by the specification for the defined shelf life of 36 months.

In Figure 17a the response surface is presented, generated using the validated neural network, while Figure 17b presents the contour graph. Contours are selected according to specification limits for the assay of 4-Hydroxy Sulphide impurity: ≤0.4%.

Based on obtained results, in Figure 17b the design space is presented; namely, the boundaries of H value of the product (10.35–10.73) in which the assay of 4-Hydroxy Sulphide impurity remains stable and in accordance with acceptance criterion prescribed by the specification for the defined shelf life of 36 months.

## 4. Conclusions

Excessive instability of Esomeprazole 40 mg solution reflects in hard-to-control pH values of a product, which primarily lays in the sensitivity of the API itself to oxygen. This lack of stability further leads to many issues such as changes in related substances’ results, affecting the assay results through the shelf life. Starting from the pH value of the bulk solution, which decreases at different speeds depending upon the conditions it is exposed to (temperature, oxygen, and time), it makes it hard to control and predict the pH value for the finished product, and consequently to control its behavior during its shelf life.

By comparing all the available results from the conducted trials, it was found that the pH level from batches kept at −30 °C is higher and more stable during the stability study than batches kept at 20 °C. This difference is a result of different oxygen exposures of the product solution after filling, prior to lyophilization, which comes from the effect that, even exposed in the same time duration, batches kept at −30 °C immediately became inert and insensitive to all potential further impacts. This is the reason why these batches with the higher pH value near the upper specification limit (specification 10.0–11.0) with a pH equal to or above 10.8 are marked as critical in comparison with batches kept at room temperature for the same time, lead to the changes in related substances results.

An MLP network, as a deep learning tool that was used in this study, provided excellent prediction and matching of experimentally obtained and predicted results and established the desired design space for the lyophilization process. The design space, described by the boundaries of pH value of 10.4–10.6 for the finished product and the prescribed shelf life of 36 months, in which levels of all related substances and assay of Esomeprazole stays in accordance with specification during whole shelf life, was developed. Keeping the pH value in this narrowed range experimentally revealed will stabilize the Esomeprazole powder for solution/injection and ensure reliable and reproducible production, meeting all predefined quality attributes and specifications for the final product.

## Figures and Tables

**Figure 1 pharmaceutics-13-00829-f001:**
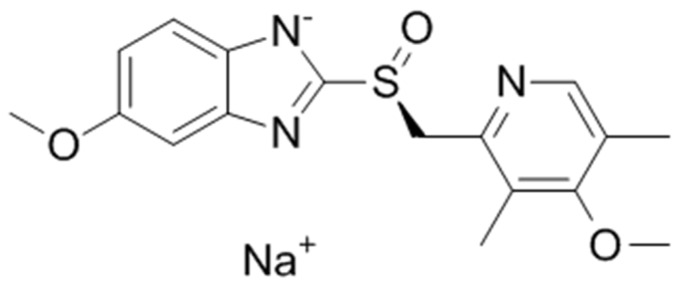
Esomeprazole sodium (sodium 5-methoxy-2-[(S)-(4-methoxy-3,5-dimethylpyridin-2-yl)methanesulfinyl]-1H-1,3-benzodiazol-1-ide).

**Figure 2 pharmaceutics-13-00829-f002:**
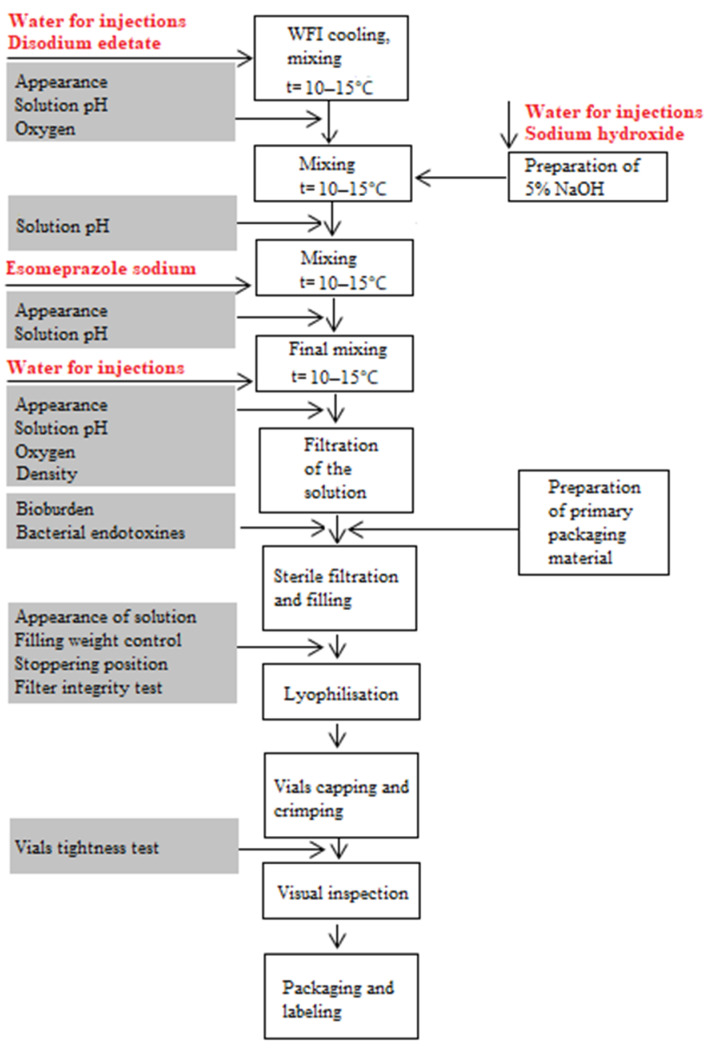
Flow chart of the manufacturing process of Esomeprazole 40 mg powder for solution for injection/ infusion. In the figure given, in red letters by the order of their addition all raw materials are presented, while details given in the white cells correspond to the description of production phases, and information in the grey cells corresponds to in-process control parameters.

**Figure 3 pharmaceutics-13-00829-f003:**
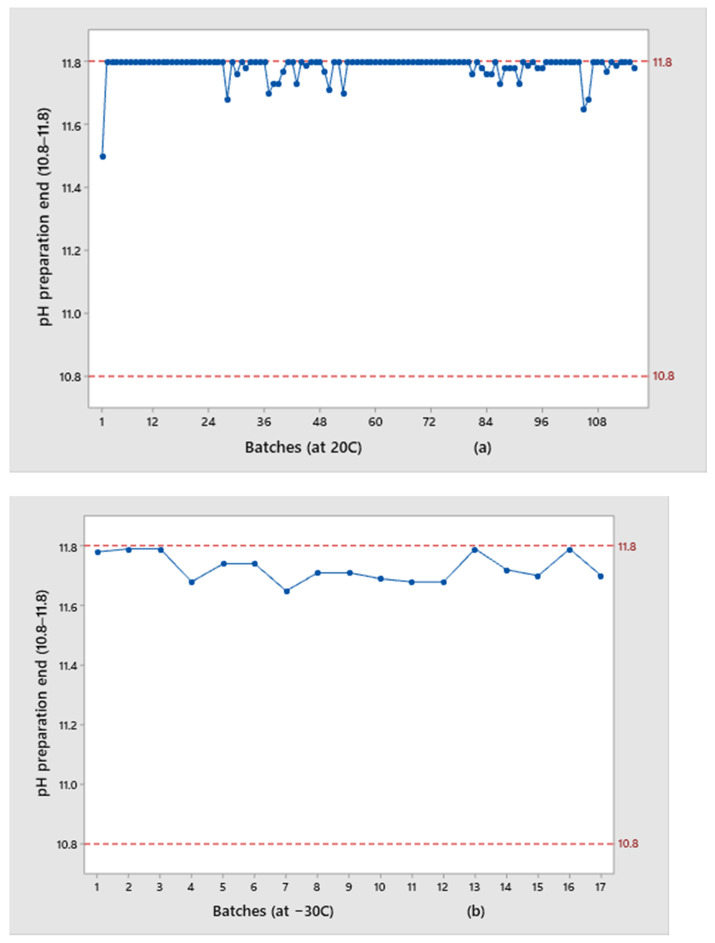
Comparison of pH values of the bulk solution at the end of preparation for the batches: (**a**) Kept at 20 °C; (**b**) Kept at −30 °C.

**Figure 4 pharmaceutics-13-00829-f004:**
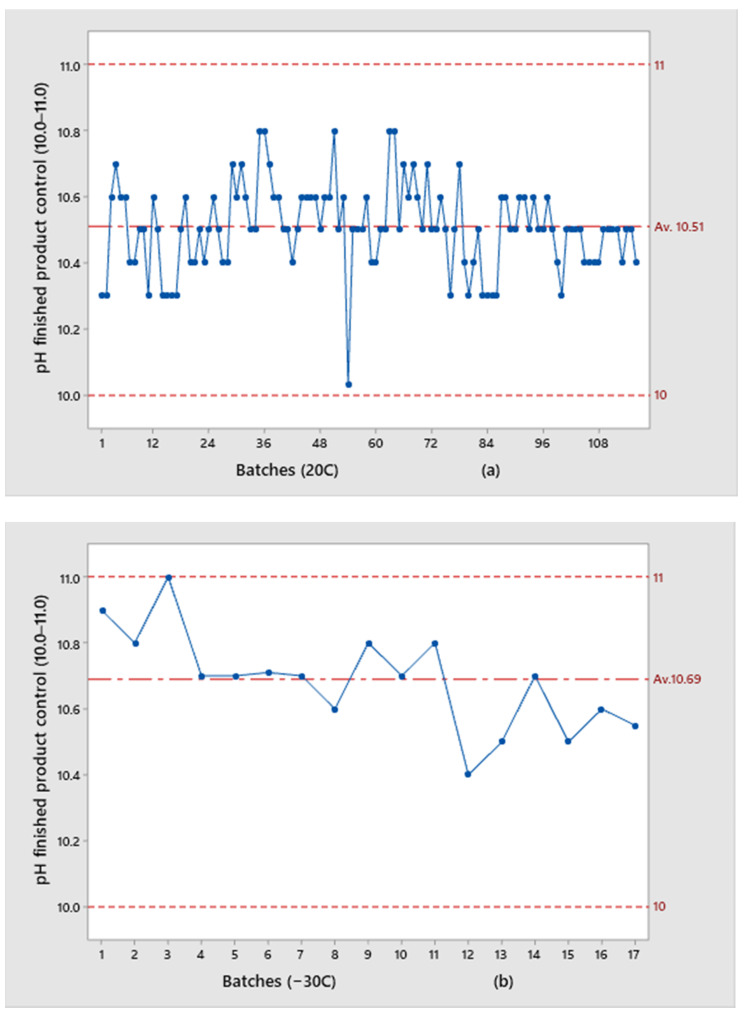
Comparison of pH values of the finished product for the batches: (**a**) kept at 20 °C; (**b**) kept at −30 °C.

**Figure 5 pharmaceutics-13-00829-f005:**
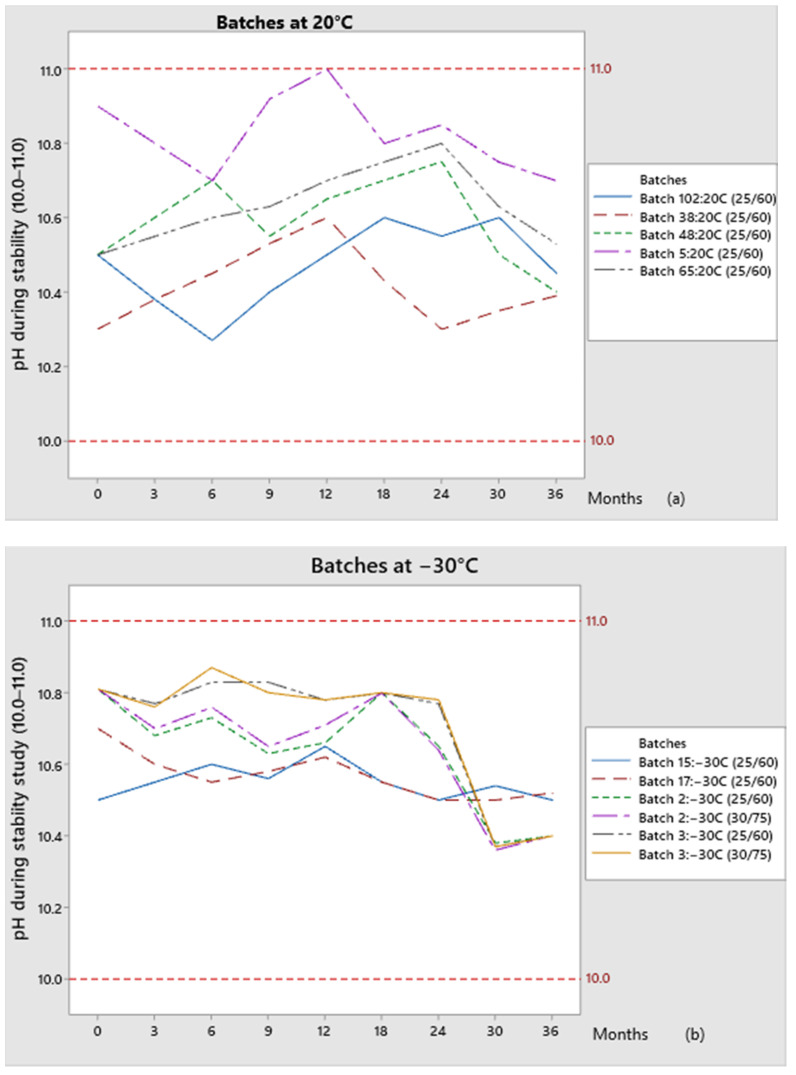
Comparison of pH values during the stability study for the batches: (**a**) Kept at 20 °C; (**b**) Kept at −30 °C.

**Figure 6 pharmaceutics-13-00829-f006:**
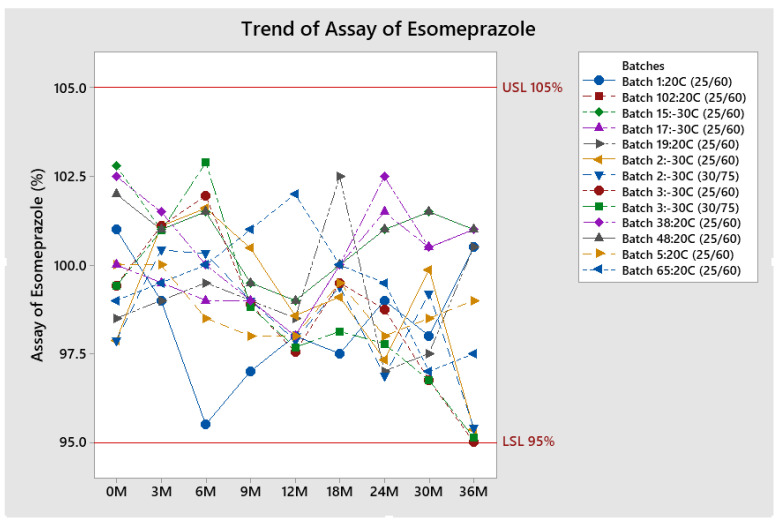
Trend of Esomeprazole results for stability batches.

**Figure 7 pharmaceutics-13-00829-f007:**
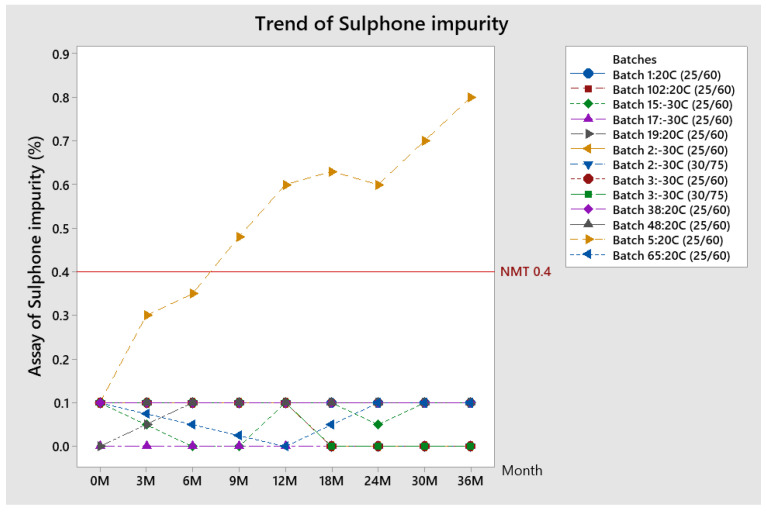
Trend of Sulphone impurity during the stability study.

**Figure 8 pharmaceutics-13-00829-f008:**
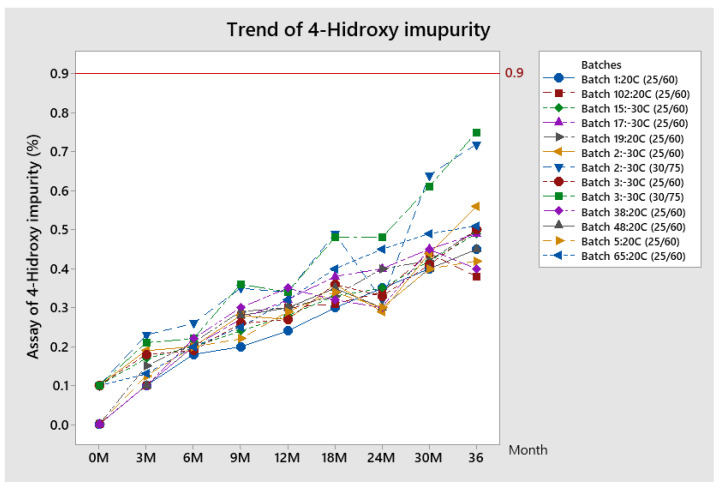
Trend of 4-hydroxy impurity for stability batches.

**Figure 9 pharmaceutics-13-00829-f009:**
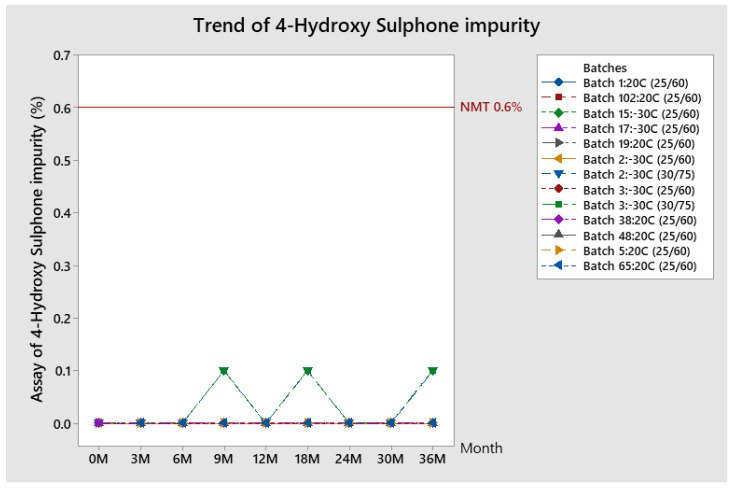
Trend of 4-hydroxy sulphone impurity for stability batches.

**Figure 10 pharmaceutics-13-00829-f010:**
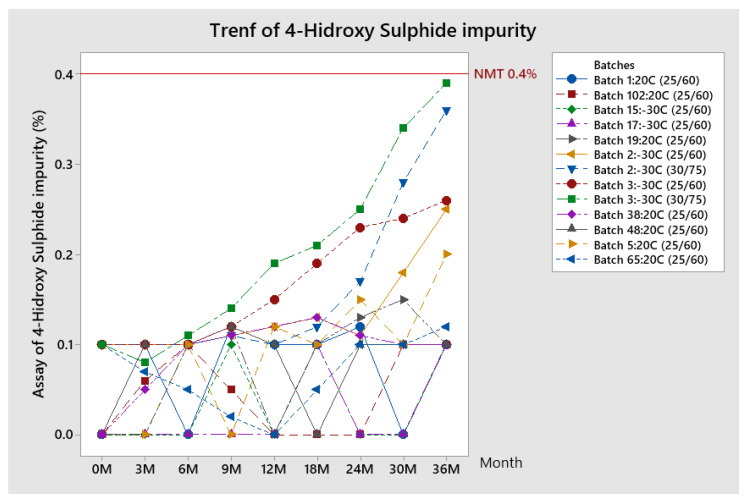
Trend of 4-hydroxy-sulfide for stability batches.

**Figure 11 pharmaceutics-13-00829-f011:**
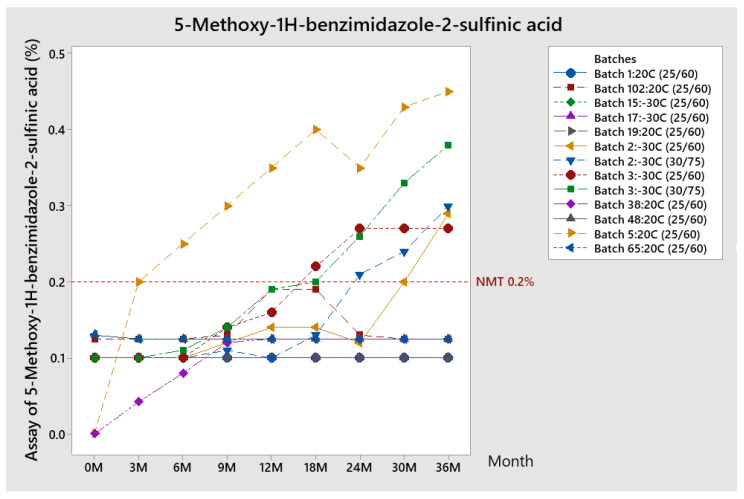
Trend of 5-Methoxy-1H-benzimidazole-2-sulfinic acid during stability study (conditions 25 °C/60% RH).

**Figure 12 pharmaceutics-13-00829-f012:**
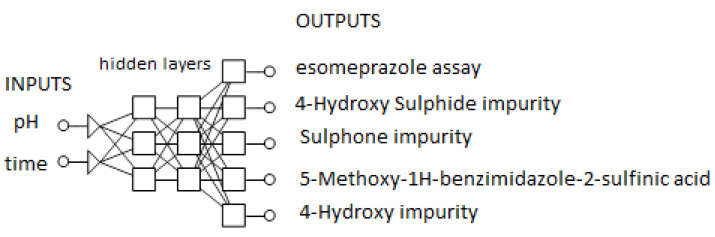
Four-layer MLP network architecture used.

**Figure 13 pharmaceutics-13-00829-f013:**
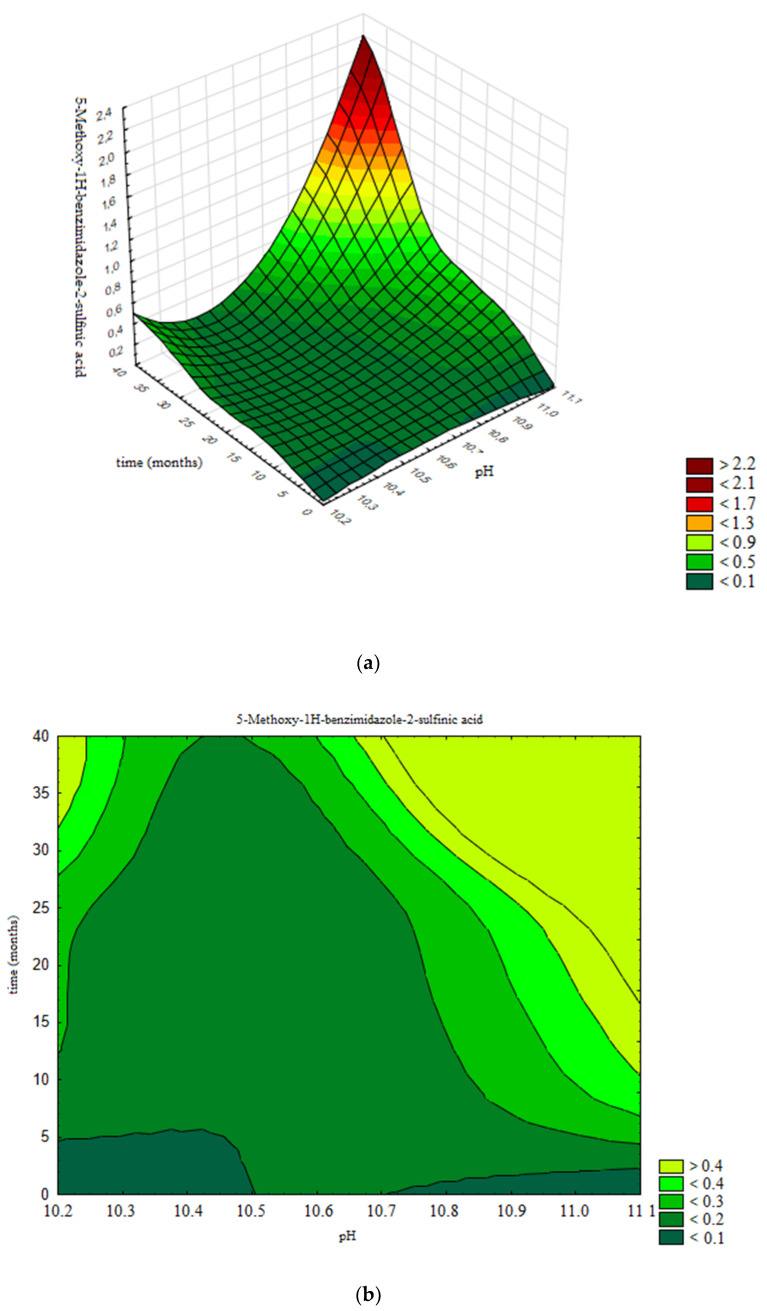
Influence of pH value and time on the assay of 5-Methoxy-1H-benzimidazole-2-sulfinic acid: (**a**) Response surface; (**b**) Contour plot.

**Figure 14 pharmaceutics-13-00829-f014:**
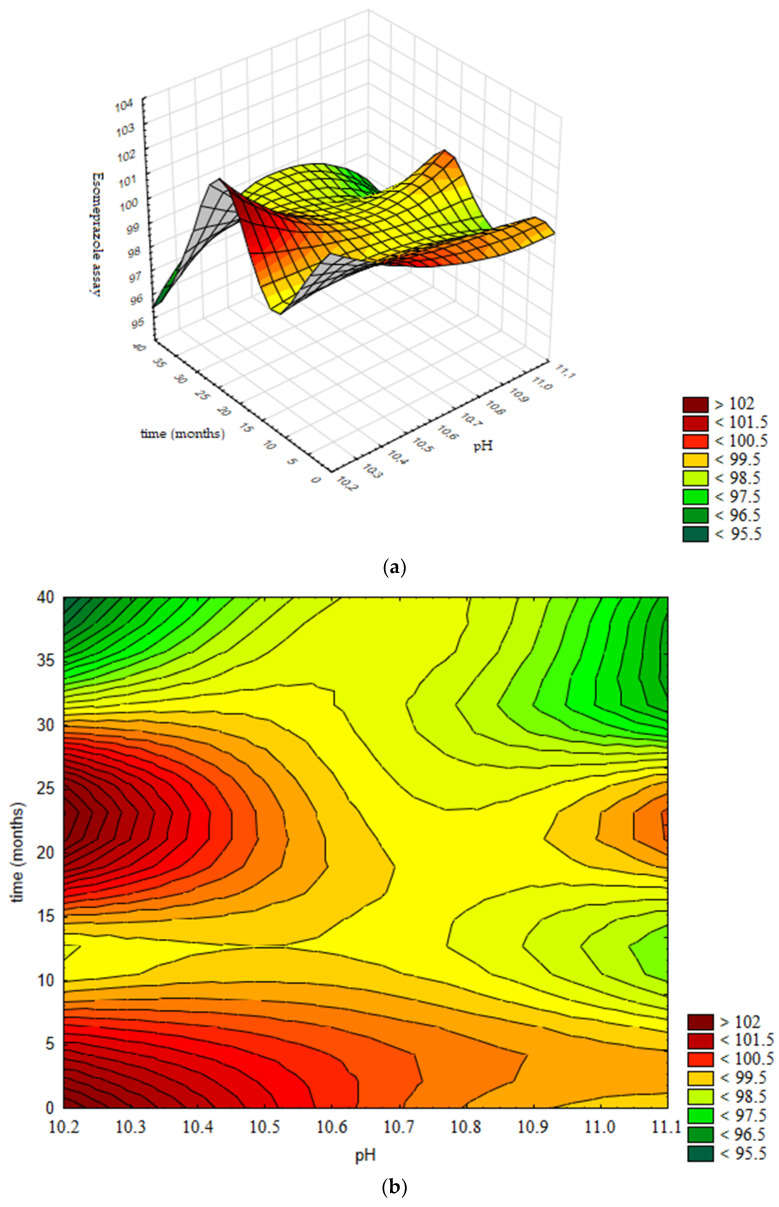
Influence of pH value and time on the assay of Esomeprazole: (**a**) Response surface; (**b**) Contour plot.

**Figure 15 pharmaceutics-13-00829-f015:**
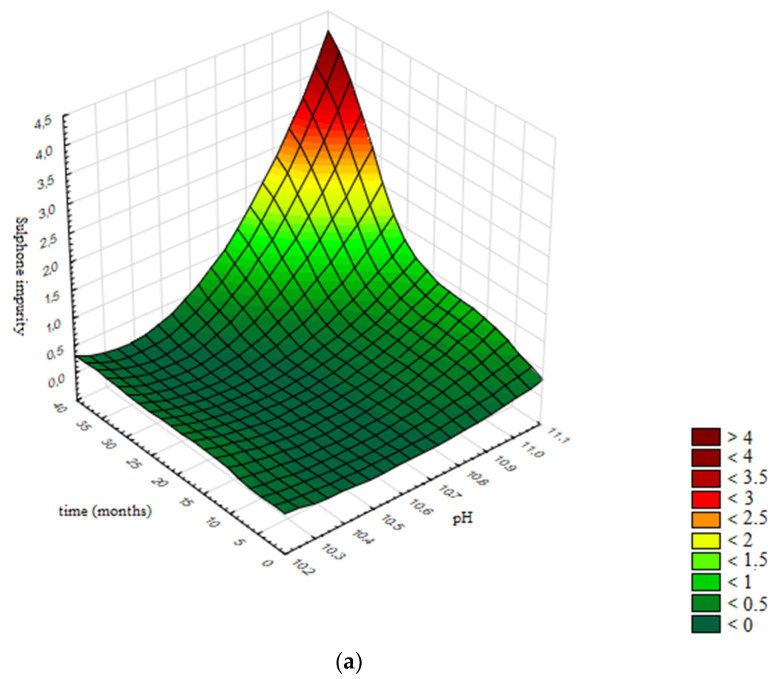
Influence of pH value and time on the assay of Sulphone impurity: (**a**) Response surface; (**b**) Contour plot.

**Figure 16 pharmaceutics-13-00829-f016:**
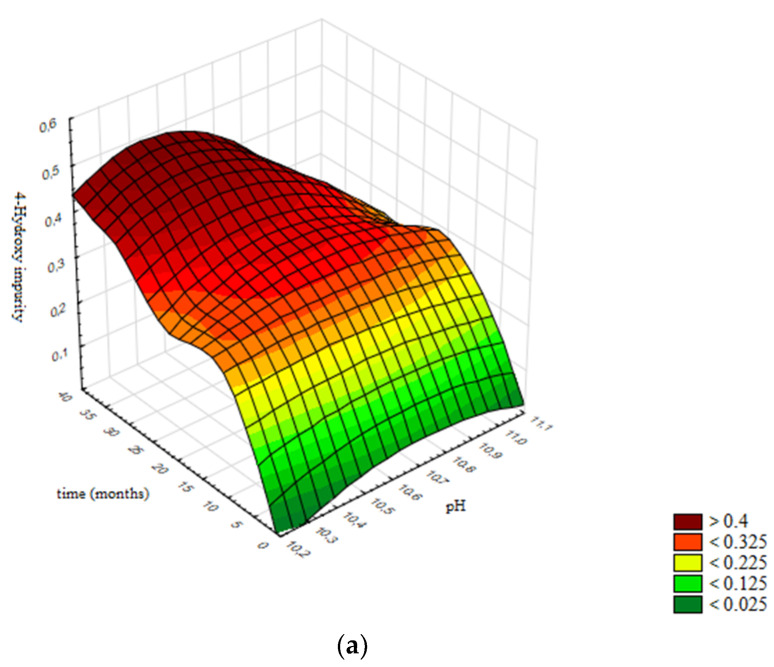
Influence of pH value and time on the assay of 4-Hydroxy impurity: (**a**) Response surface; (**b**) Contour plot.

**Figure 17 pharmaceutics-13-00829-f017:**
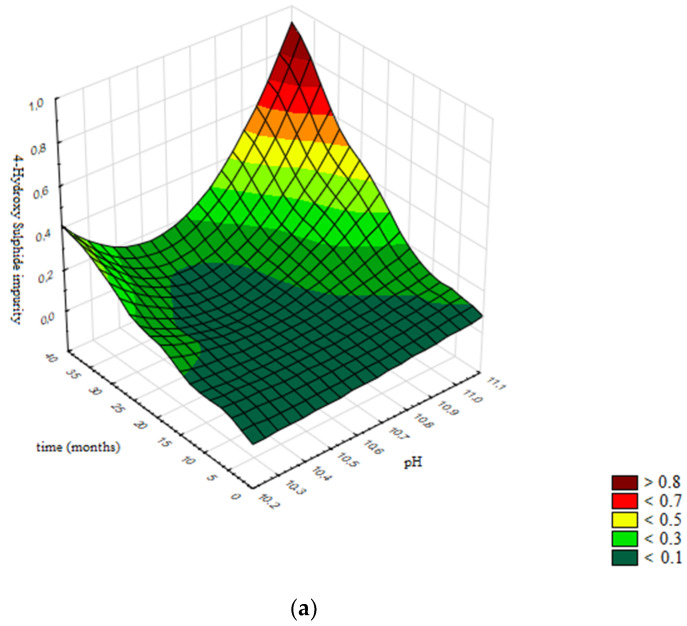
Influence of pH value and time on the assay of 4-Hydroxy Sulphide impurity: (**a**) Response surface; (**b**) Contour plot.

**Table 1 pharmaceutics-13-00829-t001:** Trials conducted for Esomeprazole 40 mg powder for solution for injection/ infusion.

Trial Type	Partially Closed Vials upon Filling Stored at a Temperature of 20 °C for a Maximum of 3 h before Starting the Lyophilization Program	Partially Closed Vials upon Filling Stored at a Temperature of −30 °C for a Maximum of 3 h before Starting the Lyophilization Program
Batch size	67.98 L 33,000 vials	67.98 L 33,000 vials
Total number of batches	115	17
Long-term stability study (25 °C/ 60% RH) for the shelf life of 36 months *	Vials of 5 batches in total	4 batches in total
Intermediate stability study (30 °C/ 75% RH) for the shelf life of 36 months *	Not applicable	2 batches in total

* Remark: Vials during the stability study were kept in the outer carton in order to protect them from light.

**Table 2 pharmaceutics-13-00829-t002:** Storage conditions and testing frequencies.

Storage Conditions		Testing Time Points (Months)
Long-term	25 ± 2 °C/60 ± 5% RH	0, 3, 6, 9, 12, 18, 24, 36
Intermediate	30 ± 2 °C/75 ± 5% RH	0, 3, 6, 9, 12, 18, 24, 36

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
