# Peer review of "Prediction of Drug Stability Using Deep Learning Approach: Case Study of Esomeprazole 40 mg Freeze-Dried Powder for Solution"

_pharmaceutics, 2021, doi:10.3390/pharmaceutics13060829_

Round 1
Reviewer 1 Report
I read the work of Ajdarić et al. on the development of metamodels able to predict the stability of esomeprazole after freeze-drying. The paper is eligible to be published after some clarifications.
- The stability of many lyophilized drugs stored for a long period can be influenced by the residual moisture at the end of secondary drying. Do the authors have the value of residual moisture in their samples? Can residual moisture affect the stability of the product?
- What are the conditions (shelf temperature, chamber pressure, duration) of freezing, primary and secondary drying?
- Freeze-drying is known to be affected by batch-to-batch and vial-to-vial variability. How did the authors account for that in analyzing their data? In fact, the quality of the data might be very important in training the ANN and, finally, having reliable metamodels.
Reviewer 2 Report
The authors applied a deep learning algorithm and modeling approach for the prediction of freeze-dried esomeprazole stability. The authors primarily evaluated the effect of pH of bulk solution on stability over storage. Overall, the manuscript is well-written, and the methods and results sections are adequate.
Here are some minor comments:
1) In Figure 2, What are the parameters highlighted in gray cells, are those the input process parameters or in-process control? It would be helpful to the readers to provide additional details regarding the manufacturing process.
2) For Figure 4, the authors mentioned "Results of pH in finish product are more shifted toward upper specification limit for batches kept at -30℃ than for those kept at 20℃" However when observed for the batches stored at-30C the pH data showed some sort of decreasing trend from batch 1 to batch 17. Batches (-30C) - leaving the first three batches, the rest of the batches (4-17) fall in the same range as the batches stored at 20C. Elaborating on this would be helpful.
3) For Figures 6, 7, 8, 9 - the legends are confusing as the long-term/intermediate storage conditions are missing for the majority of the data profiles. For example in Figure 7, what is the storage temperature for the Batch 5:20C plot? The same is the case for the rest of the figure legends where the stability info is missing.
4) Does the model incorporate inherent variability of the pH measurement?
